# Quercetin Suppresses the Progression of Atherosclerosis by Regulating MST1-Mediated Autophagy in ox-LDL-Induced RAW264.7 Macrophage Foam Cells

**DOI:** 10.3390/ijms20236093

**Published:** 2019-12-03

**Authors:** Hui Cao, Qingling Jia, Li Yan, Chuan Chen, Sanli Xing, Dingzhu Shen

**Affiliations:** Shanghai Geriatric Institute of Chinese Medicine, Shanghai University of Traditional Chinese Medicine, 365C Xiangyang South Road, Xuhui, Shanghai 200031, China; huicao19930214@gmail.com (H.C.); minibei999@gmail.com (Q.J.); adayan626@gmail.com (L.Y.); article_chenchuan@163.com (C.C.); xingsanli@163.com (S.X.)

**Keywords:** quercetin, RAW264.7, atherosclerosis, autophagy, senescence

## Abstract

Objective: To investigate the process by which quercetin suppresses atherosclerosis by upregulating MST1-mediated autophagy in RAW264.7 macrophages. Methods: An in vitro foam cell model was established by culturing RAW264.7 macrophages with oxidized low-density lipoprotein (ox-LDL). The cells were treated with quercetin alone or in combination with the autophagy inhibitor, 3-methyladenine, and autophagy agonist, rapamycin. Cell viability was detected with a CCK-8 kit. Lipid accumulation was detected by oil red O staining, senescence was detected by SA-β-gal (senescence-associated β-galactosidase) staining, reactive oxygen species were detected by ROS assay kit. Autophagosomes and mitochondria were detected by transmission electron microscope (TEM), and expression of MST1, LC3-II/I, Beclin1, Bcl-2, P21, and P16 were detected by immunofluorescence and Western blot. Results: Ox-LDL induced RAW264.7 macrophage-derived foam cell formation, reduced survival, aggravated cell lipid accumulation, and induced a senescence phenotype. This was accompanied by decreased formation of autophagosome; increased expression of P53, P21, and P16; and decreased expression of LC3-II/I and Beclin1. After intervention with quercetin, the cell survival rate was increased, and lipid accumulation and senescence phenotype were reduced. Furthermore, the expression of LC3-II/I and Beclin1 were increased, which was consistent with the ability of quercetin to promote autophagy. Ox-LDL also increased the expression of MST1, and this increase was blocked by quercetin, which provided a potential mechanism by which quercetin may protect foam cells against age-related detrimental effects. Conclusion: Quercetin can inhibit the formation of foam cells induced by ox-LDL and delay senescence. The mechanism may be related to the regulation of MST1-mediated autophagy of RAW264.7 cells.

## 1. Introduction

Cardiovascular and cerebrovascular diseases are the leading causes of morbidity and mortality worldwide, and atherosclerosis (AS) is considered to be a driver [1]. Macrophage foam cells are an important component of AS lesions and play key roles in the development of AS [2]. At the early stages of AS development, monocytes migrate to the intima to differentiate into macrophages. Macrophage phagocytosis and metabolism of oxidized low-density lipoprotein (ox-LDL) are increased, and lipids are transported from the cells to the vessel walls. When ox-LDL intake exceeds the capacity of macrophage metabolism, macrophages are transformed into foam cells, which promote the development of AS. Pulse wave velocity acts as a marker of subclinical atherosclerosis, it increases with the gradual increment of ox-LDL [3]. AS is more common in middle-aged and elderly people, and aging is an independent risk factor for AS [4]. Cell senescence can be observed in atherosclerotic plaques of AS patients [5]. Furthermore, intima–media thickness, as a risk factor for AS and a subclinical gold indicator, shows a high correlation with age [6]. All the above evidence confirms that the occurrence and development of AS are related to aging and senescence. SA-β-gal (senescence-associated β-galactosidase) staining is a widely used marker of senescence, but its activity is affected by the lysosomal state [7]. Therefore, the tumor suppressor gene P53, cyclin-dependent kinase inhibitor (P21), and multiple tumor suppressor 1 (P16) have also been used as biomarkers for the detection of cellular senescence [8]. Cellular senescence exhibits a range of morphological and physiological changes, including mitochondrial changes, chromatin condensation, etc. [9]. The aging free radical theory emphasizes the role of reactive oxygen species (ROS)-induced cell damage in aging promotion. The theory suggests that ROS are overproduced under stress conditions in cells, both natural and artificial, and are harmful to cellular components. Mitochondria are the main source of ROS and the primary target of ROS damage, and their status is closely related to the aging phenotype [10].

Autophagy, a catabolic pathway mediated by lysis, maintains the environmental balance in vivo [11]. Studies have shown that enhanced autophagy can improve age-related phenotypes, reduce age-related heart and kidney disease, and improve the health status of mice [12]. It can also protect the body from stress injury and delay the development of AS [13]. Macrophage autophagy plays an important role in the occurrence and development of AS by promoting the efflux of intracellular cholesterol. Defective autophagy of macrophages in AS impairs cholesterol metabolism, apoptosis and inflammatory body activation, and accelerates cell senescence and plaque formation [14,15]. Loss of the macrophage-specific autophagy gene Atg5 in high-fat diet-fed LDL-R^−/−^ mice increased oxidative stress, promoted plaque necrosis, and caused AS-associated thrombotic cardiovascular events [16]. Mammalian Ste20-like kinase 1 (MST1), the core component of the Hippo signaling pathway, participates in multiple biological activities of cells, including autophagy, apoptosis, and oxidative stress [17]. Furthermore, inhibition of autophagy by MST1 promotes apoptosis of macrophages, thus aggravating AS [18].

Quercetin (QUE), a natural flavonoid compound with anti-inflammatory, antiaging, and lipid metabolism regulation, plays an important role in the treatment of AS. Previous research from our group suggests that QUE can effectively reduce RAW264.7 cell damage induced by ox-LDL, improve cell viability, reduce intracellular lipid accumulation and senescence, and QUE can prevent the development of AS in apoE^−/−^ mice by regulating lipid metabolism [19,20]. QUE has also been shown to play an important role in the prevention and treatment of AS by upregulating endothelial autophagy to protect cells from stress injury [21]. Based on these observations, we explored the role of QUE in antagonizing the expression of key factors in the MST1 signaling pathway in order to provide a mechanism for the inhibition of autophagy in ox-LDL-induced RAW264.7 foam cells.

## 2. Results

### 2.1. Quercetin Increased the Viability of RAW264.7 Cells

Foam cell formation, induced by phagocytizing LDL-C and ox-LDL by macrophages in the early atherosclerotic plaques, is an important factor in the development of AS [22]. In order to establish a macrophage-derived foam cell model, we cultured RAW264.7 cells with different concentrations of ox-LDL. According to our CCK-8 assay results, cell proliferation was decreased by ox-LDL and began to level at a concentration of 100 μg/mL ox-LDL (Figure 1A). Therefore, we used 100 μg/mL ox-LDL to induce the formation of foam cells in subsequent experiments. The same concentration of ox-LDL (100 μg/mL) has been reported in other studies [23]. Next, we assessed the effect of QUE on RAW264.7 cell viability. The cell viability decreased significantly at 200 and 100 μmoL/L QUE concentrations, but 25 and 50 μmoL/L QUE caused no statistically significant decrease (Figure 1B). To study the protective effect of QUE on RAW264.7 cells treated with ox-LDL, we cotreated RAW264.7 cells with either 25 or 50 μmoL/L of QUE and 100 μg/mL of ox-LDL. The results demonstrated that 25 μmoL/L of QUE can improve the cell viability (Figure 1C). Therefore, we selected 25 μmoL/L of QUE as an optimal dose for interference with RAW264.7 cell induction by ox-LDL.

### 2.2. Quercetin Delayed Senescence and Reduced the Accumulation of Lipid in RAW264.7 Cells

Oil red O and SA-β-gal staining were used to detect the effects of QUE on lipid accumulation and senescence in RAW264.7 cells. As shown in Figure 2A,B, there was a large amount of red staining lipid accumulation in the M group (ox-LDL-treated) compared with the Con group (untreated), indicating that 100 μg/mL ox-LDL successfully induced the foam cell model. Furthermore, the addition of QUE to ox-LDL-induced foam cells (M + Q group) significantly decreased the lipid accumulation. The results of the SA-beta-gal staining assay also demonstrated that the number of positive staining cells in the M + Q group was significantly lower than that in the M group (Figure 2C,D), which confirmed these findings. We further studied the effect of QUE on the expression of P16 and P21. The results of immunofluorescence revealed more protein aggregation of P16 and P21 in the M group; however, after using QUE, the protein aggregation of P16 and P21 decreased (Figure 3A,C).The results of Western blot showed that the expression of each of these markers of senescence was increased dramatically in the M group, and that the expression in the M + Q group was significantly lower than that in the M group (Figure 3D,F). Therefore, the results suggested that QUE can effectively delay the senescence of ox-LDL-induced RAW264.7 cells and significantly reduce intracellular lipid accumulation.

### 2.3. Inhibition of Autophagy Promoted the Lipid Accumulation and Senescence of RAW264.7 Cells

Therefore, we used 3-MA (3-methyladenine) to study the role of autophagy deficiency in ox-LDL-treated RAW264.7 cells. The results demonstrated that inhibition of autophagy aggravated the lipid accumulation in ox-LDL-treated RAW264.7 cells (Figure 4A,B). Consistently, SA-β-gal staining showed more positive staining cells (Figure 4C,D), and the expression of P16 and P21 protein increased significantly (Figure 5A–F). These results suggested that inhibition of autophagy promoted lipid accumulation and senescence in RAW264.7 cells.

### 2.4. Promotion of Autophagy Inhibited the Lipid Accumulation and Senescence of RAW264.7 Cells

In addition, in this study, we used 0.1 μm autophagy agonist rapamycin to intervene in macrophages to study the role of autophagy in the treatment of AS. The results showed that intracellular lipid accumulation decreased (Figure 6A,B), and SA-β-gal staining showed senescent cells decreased (Figure 6C,D), accompanied by a decrease in P21 and P16 protein expression (Figure 7). These results suggested that importation of autophagy reduced lipid accumulation and senescence in RAW264.7 cells.

### 2.5. Quercetin Delayed Senescence, Which Was Associated with MST1-Mediated Autophagy

Next, to study the mechanism of QUE on AS, we further examined the expression of autophagy. As shown in Figure 8A, the autophagosomes were greater in the M + Q group than in the M group. Furthermore, the autophagosomes were lower in the M + 3-MA group but higher in the M + Q + 3-MA group. These results suggested that inhibition of autophagy by 3-MA can be reversed by QUE. As we all know, AS is a degenerative disease associated with aging, and autophagy can effectively delay the function of aging in AS. In the previous study, we have confirmed that QUE can promote autophagy and effectively delay aging and exert an anti-AS effect in apoE^−/−^ mice [24]. TEM showed that the mitochondria in the M group were swollen and the mitochondrial ridges were blurred. The mitochondria morphology was normal in the M + Q group, and the mitochondrial ridges were clearly visible (Figure 8B). Combined with the detection of ROS, QUE can significantly reduce the content of ROS (Figure 8C,D), confirming that QUE can effectively delay the senescence of RAW264.7 cells. We suspected that QUE acted as an anti-AS agent through autophagy and then delayed senescence. We stained the LC3 and P53, the results showed that the M + Q group had more double-labeled staining regions than the M group (Figure 9A). Due to the fact that MST1 can inhibit autophagy and aggravate AS, we speculated whether QUE can play an anti-AS role through the promotion of autophagy by inhibiting MST1. The results showed that QUE can significantly reduce the expression of MST1 and Bcl-2, and increase the expression of Beclin1 and LC3-II/I;. These findings suggested that QUE may play a role in promoting cell autophagy by inhibiting MST1.

## 3. Discussion

AS, a metabolic disease characterized by lipid deposition in the walls of the great and middle arteries, is closely related to aging [25]. As a major component of AS lesions, foam cells play an important role in the development of AS [26]. In this study, we exposed RAW264.7 cells to ox-LDL to establish a foam cell model. The cell viability was decreased, and oil red O staining showed an increase in lipid content, suggesting that the model was successfully established. The expression of P21 and P16 also increased, accompanied by an increase in SA-β-gal positive cells. As these markers are associated with cell senescence [5,26], our results suggested that RAW264.7 macrophage-derived foam cells were successfully derived and accompanied by cellular senescence.

We further demonstrated that the effects of ox-LDL on cell viability, lipid accumulation, and cell senescence in RAW264.7 cells can be partially blocked by QUE, which was consistent with our previous findings [19]. QUE is a flavonoid compound with anti-inflammatory, antioxidation, antiaging, and other cardiovascular protective effects. Previous studies have reported that QUE can reduce the expression of inflammatory factors and adhesion molecules in AS by regulating the TLR/NF-κb signaling pathway and can also upregulate the cholesterol transporter protein ABCA1 in apoE^−/−^ mice to promote cholesterol outflow, which effectively intervenes with AS [27,28]. In the foam cell model induced by ox-LDL, QUE can effectively upregulate the expression of ABCA1 and activator of transcription PPAR, promote the RCT pathway, reduce the formation of foam cells, and further prevent AS [29]. Our findings demonstrated that QUE can effectively reduce the accumulation of intracellular lipid and delay cell senescence, which further confirmed the anti-AS and aging effect of QUE.

Autophagy promotes cellular homeostasis and stress adaptation, and plays an important role in the occurrence and development of AS. Studies have shown that autophagy inhibitor 3-MA significantly increases accumulation of lipid in THP-1 cells. Inhibition of autophagy accelerates cell senescence, promotes AS development, and accelerates the formation of plaque [5,15]. As a widely used autophagy agonist, RAP can promote autophagy, mainly by inhibiting the mTOR (Mammalian rapamycin target protein) signaling pathway. Studies have shown that RAP can upregulate autophagy of macrophages, reduce the accumulation of lipid, and delay senescence, which plays an effective role in anti-AS activity [30,31]. Our results also confirmed that 3-MA promoted the accumulation of intracellular lipid and aggravated senescence, while RAP significantly reduced the accumulation of intracellular lipid and reduced the expression of senescence protein.

Recently, it has been demonstrated that QUE can upregulate autophagy in endothelial cells through the ERK pathway, thus effectively preventing AS [20]. Autophagy, a subcellular process by which lysosomes degrade damaged organelles and proteins, plays a key role in the development of AS. Inducing macrophage autophagy can reduce the aggregation of macrophage in AS plaques, protect cells from inflammatory factors, and promote the stability of plaque [32]. Autophagy can also promote intracellular lipid hydrolysis and cholesterol efflux in macrophage-derived foam cells, thereby effectively inhibiting the progression of AS [33]. Our results demonstrated that the number of autophagosomes decreased after exposure of RAW264.7 cells to ox-LDL, and that decrease was blocked by QUE. Furthermore, QUE promoted the expression of LC3-II/Ⅰ, reversed the AS-promoting effect of the autophagy inhibitor 3-MA, and reduced the expression of P21 and P16 proteins. Mitochondria are closely related to aging and the morphology of mitochondria change with the aging process. Typical mitochondria have well-stratified mitochondrial mites, while in senescent cells, mitochondrial mites are unclear and appear as irregular vesicles [34]. ROS are mainly derived from mitochondria, and low levels of ROS are associated with delayed biosenescence [35]. Our TEM showed that the mitochondria in the M + Q group was clearly visible compared with the M group, the mitochondrial morphology was normal, and autophagosomes were produced. The results of ROS testing showed that QUE can reduce the production of ROS in cells, suggesting that it can effectively delay aging. These findings further supported the role of QUE-induced autophagy in protecting macrophages against the detrimental effects associated with AS. Therefore, these findings supported an anti-AS role for QUE by promoting macrophage autophagy and delaying senescence.

As a potential mechanism for QUE-associated effects on autophagy, we also examined effects on the expression of MST1, a serine-threonine kinase involved in a variety of biological functions, including autophagy, apoptosis, and oxidative stress [17]. Our results demonstrated that QUE reduced the expression of MST1 in ox-LDL-induced RAW264.7 foam cells, which was consistent with the possibility that QUE may promote autophagy by inhibiting the expression of MST1. MST1 has been reported to phosphorylate Beclin1 at threonine 108 in the BH3 domain, which enhances its interaction with Bcl-2/Bcl-xl, inhibits the PI3 kinase activity of the Atg14L-Beclin1-Vps34 complex, and inhibits autophagy [36]. In a diabetic mouse model, knockout of MST1 significantly upregulated autophagy and the expression of LC3-II in cardiac microvascular endothelial cells increased [16]. In recent years, MST1 has been shown to be closely related to cardiovascular and metabolic diseases [37]. Knockout of MST1 in apoE^−/−^ mice reduced the plaque area, lipid core, and macrophage accumulation. Conversely, apoE^−/−^ mice overexpressing MST1 showed a large accumulation of lipid core and macrophages. Additionally, MST1 knockout mice had significantly increased LC3-II expression and autophagosome numbers in macrophages [38]. Therefore, results from previous studies overwhelmingly support its role in autophagy and are consistent with our findings.

In summary, our results demonstrate that QUE effectively reduces ox-LDL-induced RAW264.7 foam cell formation, reducing cellular lipid accumulation and delaying cell senescence. This is associated with increased autophagy, as evidenced by an increased number of autophagosomes in the QUE group, increased LC3-II/Ⅰ, Beclin1 protein expression, and downregulation of MST1, Bcl-2, P21, and P16 expression. The above results are consistent with a mechanism for QUE in inhibiting AS by enhancing autophagy and delaying senescence through the MST1 pathway.

## 4. Materials and Methods

### 4.1. Cells and Reagents

RAW264.7 mouse macrophages were obtained from the cell bank of the Chinese Academy of Sciences. The following reagents were also used: Dulbecco’s modified Eagle’s medium (DMEM) medium (Gibco, 11965-092, Waltham, MA, USA), fetal bovine serum (FBS, Invitrogen, 10099-141, Carlsbad, CA, USA), quercetin (Shanghai Yuanye Biotechnology Co., Ltd., B20527, Shanghai, China), ox-LDL (Shanghai Yuanye Biotechnology Co., Ltd., S24879), 3-methyladenine (Sigma, M9281, St. Louis, MI, USA),rapamycin (Sigma, V900930), anti-rabbit MST1 (Cell Signaling Technology, 14946, Boston, MA, USA), anti-rabbit LC3A/B (Cell Signaling Technology, 4108), anti-rabbit Bcl-2(Abcam, ab182858, Cambridge, UK), anti-rabbit Beclin1 (Abcam, ab210498), anti-mouse P53 (Abcam, ab26), anti-rabbit P21 (Abcam, ab188224), anti-rabbit P16 (Abcam, ab51243), anti-mouse IgG (Cell Signaling Technology), anti-rabbit IgG (Cell Signaling Technology, 7074 P2), protein ladder (Thermo Fisher, 26616, Waltham, MA, USA), DAPI Staining Solution (Shanghai Beyotime Biotechnology Co., Ltd., C1006, Shanghai, China), BCA protein assay kit (Shanghai Beyotime Biotechnology Co., Ltd., P0010), SDS-PAGE Gel Preparation Kit (Shanghai Beyotime Biotechnology Co., Ltd., P0012A), Oil red O Staining Kit (Shanghai Yi Sheng Biotechnology Co., Ltd., 40759, Shanghai, China), Reactive Oxygen Species Assay Kit (Shanghai Beyotime Biotechnology Co., Ltd., S0033), In Situ β-galactosidase Staining Kit (Shanghai Beyotime Biotechnology Co., Ltd., RG0039), and Cell Counting Kit-8 (Shanghai Beyotime Biotechnology Co., Ltd., C0038).

### 4.2. Cell Culture

RAW264.7 macrophages were cultured in DMEM containing 10% FBS at 37 °C in a 5% CO_2_ incubator at constant temperature. After growth as a dense monolayer, the cells were routinely passed to a third generation. After 8 h of starvation in serum-free DMEM, the cells were randomly divided into different groups: Con (control); M (model); Q (quercetin); 3-MA (3-methyladenine), Rap (rapamycin); M + Q; M + 3-MA; M + Rap; and M + Q + 3-MA.

### 4.3. Cell Proliferation Assay

When the cells reached 70–80% confluency, they were lifted from the plates by trypsinization and diluted to 50,000/mL in culture medium. The cells were inoculated in 96-well plate with 100 μL cell suspension per well. After cell adherence, the corresponding drugs were added for 24 h. Then, 10 μL CCK-8 solution was added per well, and the absorbance optical density (OD) at 450 nm wavelength was measured in an enzyme labeling instrument after 1 h.

### 4.4. Oil Red O Staining of Cells

RAW264.7 cells were cultured in 24-well sterile culture plates and treated with corresponding drugs. The cells were gently rinsed twice with PBS solution and fixed with 4% paraformaldehyde for 30 min. The PBS solution was rinsed for 1 min, oil red O working fluid was applied in an oven at 60 °C for 15–20 min, and the cells were washed with distilled water 1–2 times, 1–2 min each time. Subsequently, 60% isopropanol was separated and the red intracellular lipid droplets were observed under the microscope. The images were collected and the experimental results were analyzed by Image J analysis software. The intracellular lipid droplets were expressed by integrated optical density.

### 4.5. SA-β-gal Staining of Cells

RAW264.7 cells were cultured in 24-well sterile culture plates and treated with corresponding drugs. Then, the culture medium was discarded and the cells were fixed with 0.5 mL beta-gal staining stationary solution for 15 min. The cells were incubated with 0.5–1 mL SA-β-gal staining working fluid. A minimum of 200 cells were observed by optical microscopy, and the percentage of positive staining cells was calculated.

### 4.6. Transmission Electron Microscopy (TEM) Observation of Autophagosome and Mitochondria

After discarding the culture medium, 2.5% glutaraldehyde was quickly added to the cultured cells. The cells were scraped gently and collected into centrifuge tubes. After discarding the glutaraldehyde, the cells were centrifuged for 5 min at 1000 rpm. A new batch of glutaraldehyde was added, and then the cells were fixed in 2% OSO4. Graded alcohol was dehydrated, resin was embedded, and ultrathin sections were stained with uranyl acetate and citric acid. Autophagosomes and cellular mitochondria were observed by transmission electron microscopy (JEM-1230, Tokyo, Japan).

### 4.7. Reactive Oxygen Species Assay of Cells

RAW264.7 cells were cultured in 24-well sterile culture plates and treated with corresponding drugs. DCFH-DA was diluted 1:1000 with serum-free medium to a final concentration of 10 μmoL/L. The cell culture medium was removed and an appropriate volume of diluted DCFH-DA was added and incubated in a 37 °C cell incubator for 20 min. The cells were washed 3 times with serum-free cell culture medium to fully remove DCFH-DA that had not entered the cells. They were then observed by fluorescence microscope.

### 4.8. Immunofluorescence Detection of Protein Expression in Macrophages

RAW264.7 cells were inoculated into 24-well culture plates with aseptic cover slides and treated with corresponding drugs. After rinsing in PBS, the cells were fixed in 4% paraformaldehyde for 30 min, rinsed 3 times for 5 min with PBS, and 0.1% Triton-X-100 (PBST) statically placed at room temperature for 15 min. The cells were blocked with 5% BSA for 1 h, after which the blocking solution was discarded, primary antibody was added, and the cells were incubated overnight at 4 °C. The cells were then rinsed 3 times for 5 min in PBST and incubated with fluorescent secondary antibody. After incubation at room temperature for 1 h in the dark, the cells were stained with DAPI for 3 min in the dark, washed 3 times for 5 min with PBST, and then sealed and observed under a laser scanning confocal microscope.

### 4.9. Western Blot Detection of MST1, LC3II/I, Beclin1, Bcl-2, P21, and P16 Expression in Macrophages

RAW264.7 cells were cultured in 6-well sterile culture plates and treated in groups. The culture medium was discarded, and RIPA lysis buffer containing PMSF was added. The cells were scraped gently and collected into centrifuge tubes. After centrifugation at 12,000 rpm, 4 °C, for 30 min, the supernatants were collected for protein measurement using the BCA method. Samples were mixed with 5× loading buffer and heated in boiling water for 10 min to denature proteins. Samples were resolved in SDS-PAGE gels and then transferred to PVDF membranes. The membranes were blocked with 5% skim milk and then incubated in corresponding antibody solutions (all diluted at 1:1000) overnight. After washing, the membranes were incubated at room temperature with secondary antibodies for 1 h. Protein band images were acquired and analyzed as the integrated absorbance (IA = mean OD × area) using Image-J software, and the relative levels of target proteins were normalized to Gapdh (target protein IA/Gapdh IA).

### 4.10. Statistical Analysis

Statistical analysis was conducted using SPSS23.0 software, and figures were generated using Graph Pad Prism 5 Project software and Adobe illustrator CC. Results were presented as means ± standard deviation (SD). Differences between groups were determined by nonpaired tests or one-way ANOVA. Differences at *p* < 0.05 were considered significant.

## 5. Conclusions

In summary, our results demonstrate that QUE effectively reduces ox-LDL-induced RAW264.7 foam cell formation, reducing cellular lipid accumulation and delaying cell senescence. This is associated with increased autophagy, as evidenced by the increased number of autophagosomes in the quercetin group; increased LC3; Beclin1 protein expression; and downregulation of MST1, Bcl-2, P21, and P16 expression. The above results are consistent with a mechanism for QUE in inhibiting AS by enhancing autophagy and delaying senescence through the MST1 pathway.

## Figures and Tables

**Figure 1 ijms-20-06093-f001:**
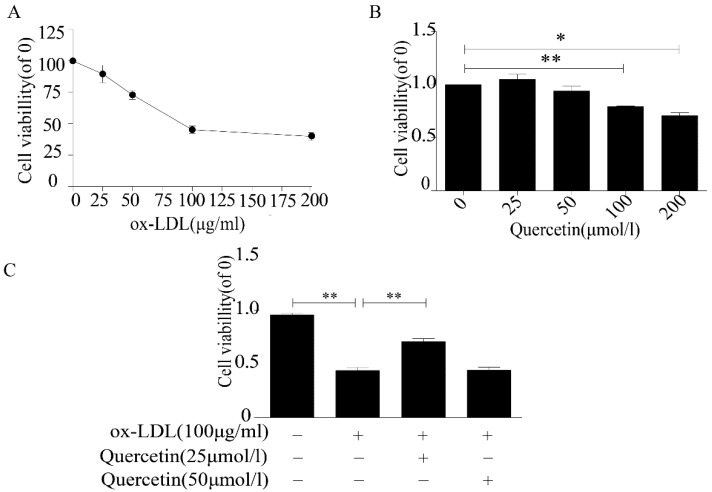
Quercetin increased the viability of RAW264.7 cells treated with oxidized low-density lipoprotein (ox-LDL). (**A**) viability of RAW264.7 cells treated with ox-LDL. (**B**) Viability of RAW264.7 cells treated with quercetin. (**C**) Viability of RAW264.7 cells treated with quercetin and ox-LDL. Data are presented as means ± SD, * *p* < 0.05; ** *p* < 0.01.

**Figure 2 ijms-20-06093-f002:**
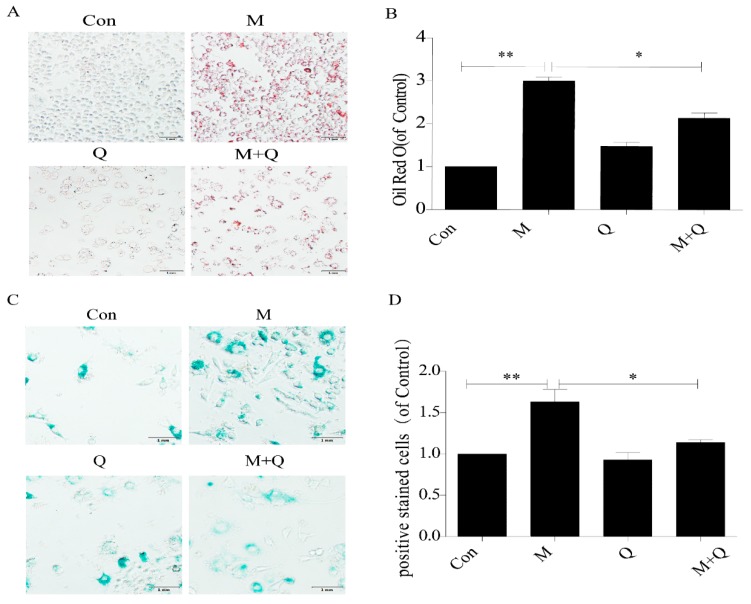
Quercetin can delay senescence of RAW264.7 cells and reduce the accumulation of intracellular lipid. (**A**) Oil red O staining. (**B**) Intracellular lipid deposition. (**C**) SA-β-gal staining. (**D**) Percentage of SA-β-gal positive stained cells. Con, control; M, model; Q, quercetin; M + Q, model + quercetin. Data are presented as means ± SD, * *p* < 0.05; ** *p* < 0.01.

**Figure 3 ijms-20-06093-f003:**
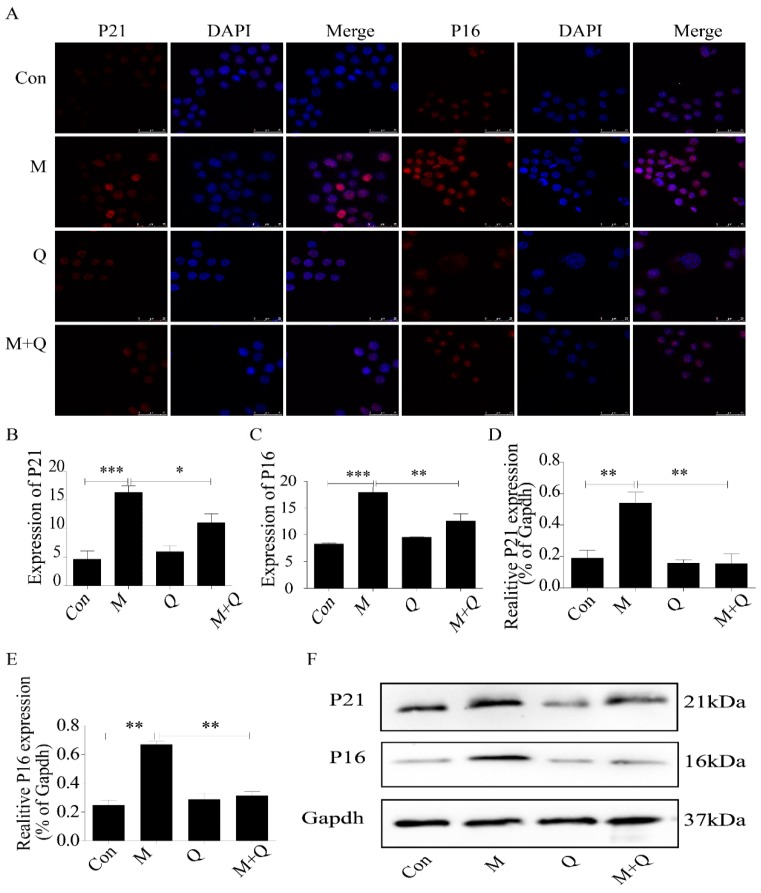
Expression of P21 and P16 in macrophage cells detected by immunofluorescence and Western blot. (**A**) Immunofluorescence. (**B**,**C**) Results of P21 and P16 immunofluorescence. (**D**,**E**) Results of P21 and P16 Western blot. (**F**) Western blot. Con, control; M, model; Q, quercetin; M + Q, model + quercetin. Data are presented as means ± SD, * *p* < 0.05; ** *p* < 0.01; *** *p* < 0.001.

**Figure 4 ijms-20-06093-f004:**
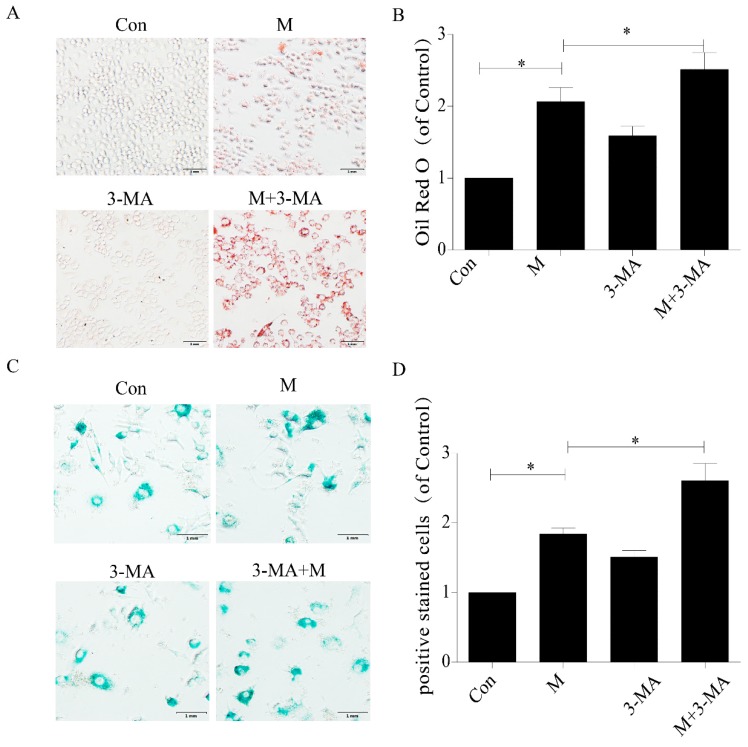
3-MA promoted senescence of RAW264.7 cells and aggravated accumulation of intracellular lipid. (**A**) Oil red O staining. (**B**) Intracellular lipid deposition. (**C**) SA-β-gal staining. (**D**) Percentage of SA-β-gal-positive stained cells. Con, control; M, model; 3-MA, 3-methyladenine; M + 3-MA, model + 3-methyladenine. Data are presented as means ± SD, * *p* < 0.05.

**Figure 5 ijms-20-06093-f005:**
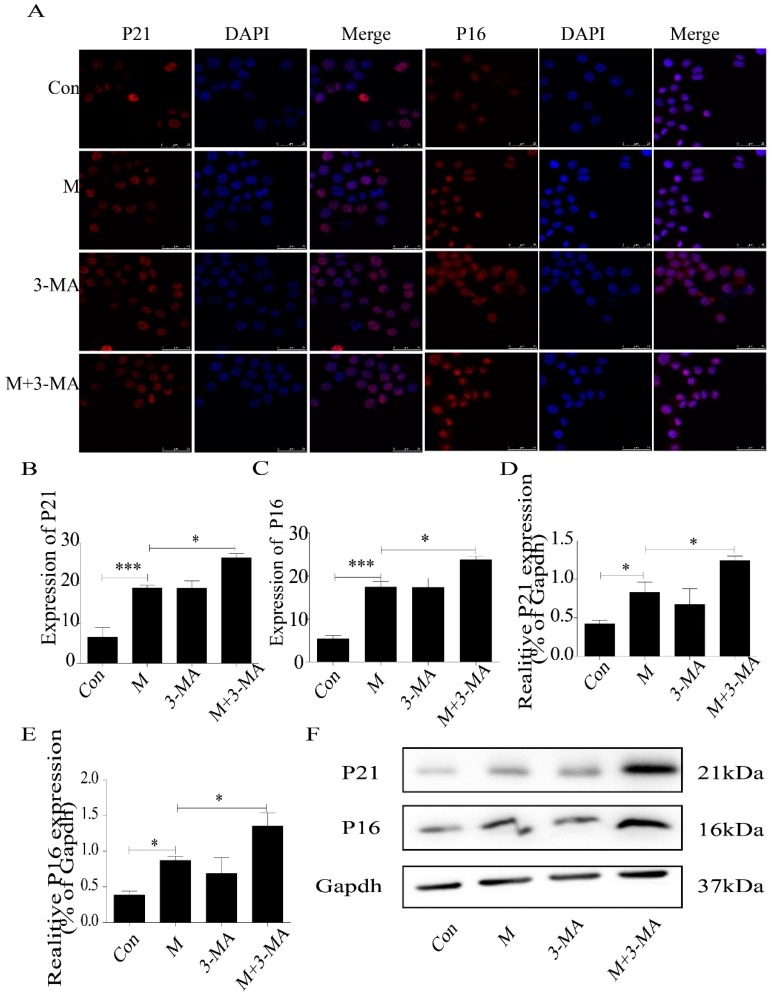
Expression of P21 and P16 in macrophage cells detected by immunofluorescence and Western blot. (**A**) Immunofluorescence. (**B**,**C**) Results of P21 and P16 immunofluorescence. (**D**,**E**) Results of P21 and P16 Western blot. (**F**) Western blot. Con, control; M, model; 3-MA, 3-methyladenine; M + 3-MA, model + 3-Methyladenine. Data are presented as means ± SD, * *p* < 0.05; *** *p* < 0.001.

**Figure 6 ijms-20-06093-f006:**
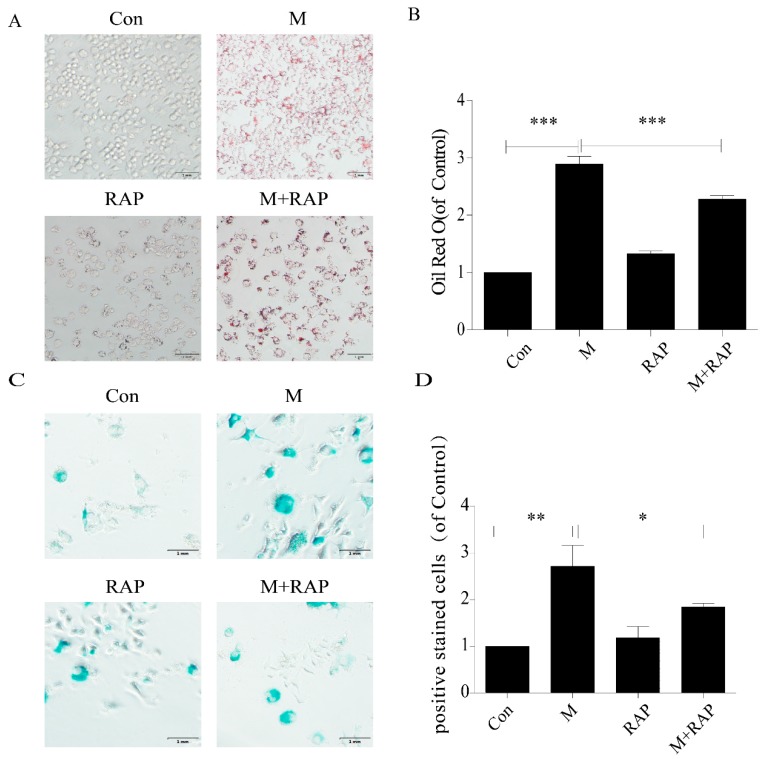
Rapamycin delayed senescence of RAW264.7 cells and reduced accumulation of intracellular lipid. (**A**) Oil red O staining. (**B**) Intracellular lipid deposition. (**C**) SA-β-gal staining. (**D**) Percentage of SA-β-gal-positive stained cells. Con, control; M, model; RAP, rapamycin; M + RAP, model + rapamycin. Data are presented as means ± SD, * *p* < 0.05; ** *p* < 0.01; *** *p* < 0.001.

**Figure 7 ijms-20-06093-f007:**
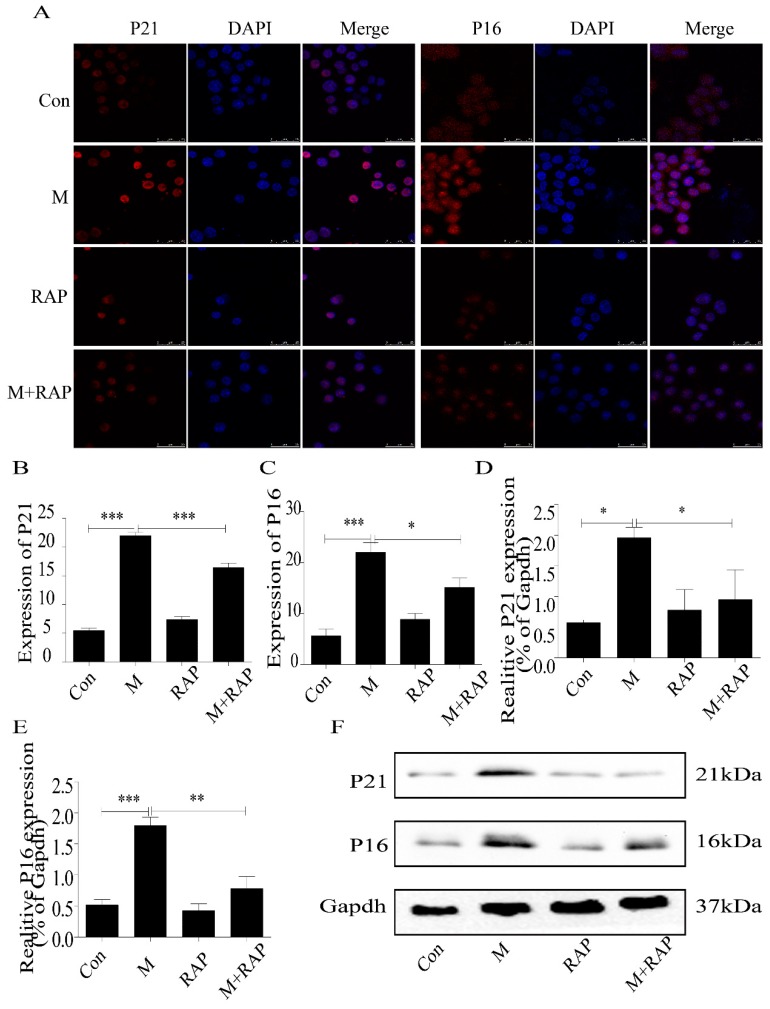
Expression of P21 and P16 in macrophage cells detected by immunofluorescence and Western blot. (**A**) Immunofluorescence. (**B**,**C**) Results of P21 and P16 immunofluorescence. (**D**,**E**) Results of P21 and P16 Western blot. (**F**) Western blot. Con, control; M, model; RAP, rapamycin; M + RAP, model + rapamycin. Data are presented as means ± SD, * *p* < 0.05; ** *p* < 0.01; *** *p* < 0.001.

**Figure 8 ijms-20-06093-f008:**
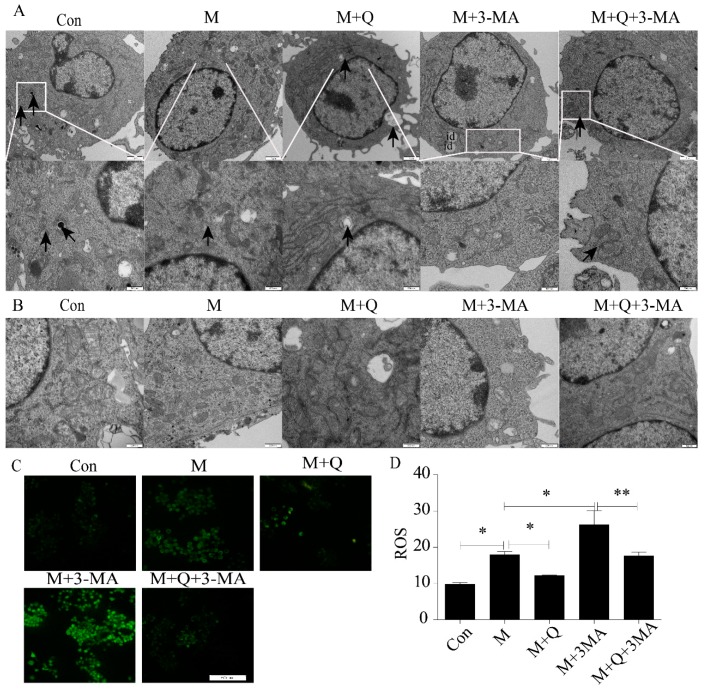
Quercetin delayed senescence by upregulating autophagy. (**A**) Autophagosomes labeled by arrow. (**B**) Cellular ultrastructure. (**C**) ROS. (**D**) Results of ROS, expressed by the average optical density value. id, lipid droplet. Con, control; M, model; M + Q; model + quercetin; M + 3-MA, model + 3-methyladenine; M + Q + 3-MA, model + quercetin + 3-methyadenine. Data are presented as means ± SD, * *p* < 0.05; ** *p* < 0.01.

**Figure 9 ijms-20-06093-f009:**
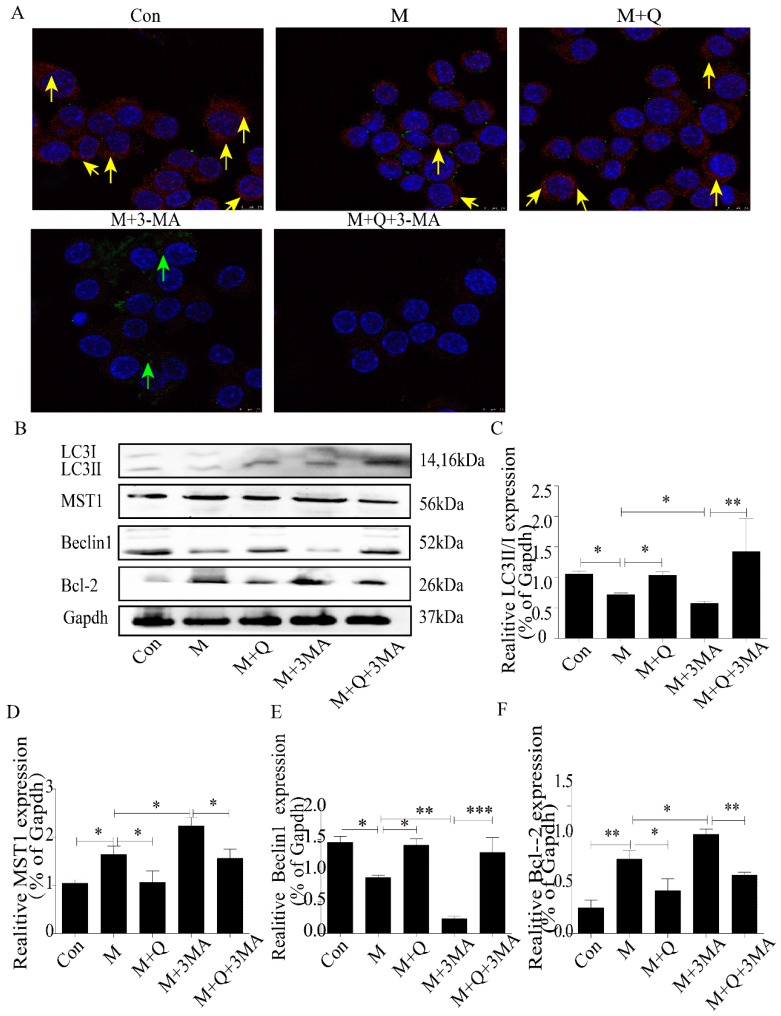
Quercetin regulated MST1-mediated autophagy. (**A**) Immunofluorescence of LC3 and P53, colocalization of P53 and LC3 labeled by yellow arrow, green arrows represented green fluorescent P53. (**B**) Western blot. (**C**–**F**) Expression of LC3-Ⅱ/Ⅰ, MST1, Beclin1, and Bcl-2. Con, control; M, model; M + Q; Model + Quercetin; M + 3-MA, model + 3-Methyladenine; M + Q + 3-MA, model + quercetin + 3-methyadenine. Data are presented as means ± SD, * *p* < 0.05; ** *p* < 0.01; *** *p* < 0.001.

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
