# Peer review of "Quercetin Suppresses the Progression of Atherosclerosis by Regulating MST1-Mediated Autophagy in ox-LDL-Induced RAW264.7 Macrophage Foam Cells"

_ijms, 2019, doi:10.3390/ijms20236093_

Round 1

Reviewer 1 Report

The manuscript is well written and can be accepted for publication. Few points need to be addressed.

Please explain the abbreviations used in the figures in their figure legends. What is the meaning of "2.1 Subsection". Why this caption is used? In the section "In order to establish a macrophage-derived .......different concentrations of ox-LDL.", please mention the concentrations used and why the authors choose this model (any previous reference).

Author Response

Dear reviewer:

Thank you very much for your comments concerning our manuscript entitled “Quercetin suppresses the progression of atherosclerosis by regulating MST1-mediated autophagy in ox-LDL-induced RAW264.7 macrophage foam cells”. Those comments are all valuable and very helpful for revising and improving our paper, as well as the important guiding significance to our researches. We have studied comments carefully and have made correction.

Point 1. Please explain the abbreviations used in the figures in their figure legends.

Response1.We have carefully studied the comments and suggestions of the reviewers and marked the abbreviations used in the pictures. Thank you very much for your comments.

Point2. What is the meaning of "2.1 Subsection". Why this caption is used?

Response2. Thank you very much for your comments. The 2.1 subsection is the content in the template when the article was submitted. Because of personal mistakes, it has not been modified according to our manuscript. Thank you very much for pointing out the problem and we made changes in time.

Point3. "In order to establish a macrophage-derived .......different concentrations of ox-LDL.", please mention the concentrations used and why the authors choose this model (any previous reference).

Respones3. Macrophage foam cells are an important component of AS lesions and play key roles in the development of AS. At the early stages of AS development, monocytes migrate to the intima to differentiate into macrophages. Macrophage phagocytosis and metabolism of oxidized low density lipoprotein (ox-LDL) are increased, and lipids are transported from the cells to the vessel walls. When ox-LDL intake exceeds the macrophage metabolic capacity, macrophages are transformed into foam cells, which promote the development of AS. Therefore, we use ox-LDL to interfere macrophages to establish a foam cell model. The concentration of ox-LDL we use is 100 μg / ml, which is also used in related studies. We have added corresponding explanations and references. Thank you very much for your comments.

Reviewer 2 Report

Dear Editor,

I carefully read the manuscript ijms-649984, which regards findings from an in vitro foam cell model. The study is interesting and of potential interest for the readers of the IJMS. I have some comments for the authors:

Reference 1 is not adequate. As a matter of fact, the authors refer to an epidemiological datum by citing a pre-clinical study. Please, replace it with doi: 10.1097/HJH.0000000000001927. In the Introduction, please add and comment the reference doi: 10.1016/j.ijcard.2018.03.077 referring to interaction among oxLDL-C and pulse wave velocity, which is a marker of sub-clinical atherosclerosis.  English language needs to be carefully revised by a native English-speaking person, in order to correct the typos.

Author Response

Dear reviewer:

Thank you very much for your comments concerning our manuscript entitled “Quercetin suppresses the progression of atherosclerosis by regulating MST1-mediated autophagy in ox-LDL-induced RAW264.7 macrophage foam cells”. Those comments are all valuable and very helpful for revising and improving our paper, as well as the important guiding significance to our researches. We have studied comments carefully and have made correction. We read Reference 1 carefully and found that it is not very appropriate to support the epidemiological data by citing preclinical research literature, so we refer to the reviewer's suggestion to use doi: 10.1097 / HJH.0000000000001927 instead, and we added References doi: 10.1016 / j.ijcard.2018.03.077 in the introduction. We have also carefully revised the language. Thank you very much for your comments.